# FEDERATED TEXT-DRIVEN PROMPT GENERATION FOR VISION-LANGUAGE MODELS

**Chen Qiu**[*]
Bosch Center for AI, USA

**Xingyu Li**[*†]
Tulane University

**Chaithanya Kumar Mummadi & Madan Ravi Ganesh & Zhenzhen Li**
Bosch Center for AI, USA

**Lu Peng**
Tulane University

**Wan-Yi Lin**
Bosch Center for AI, USA

## ABSTRACT

Prompt learning for vision-language models, e.g., CoOp, has shown great success in adapting CLIP to different downstream tasks, making it a promising solution for federated learning due to computational reasons. Existing prompt learning techniques replace hand-crafted text prompts with learned vectors that offer improvements on seen classes, but struggle to generalize to unseen classes. Our work addresses this challenge by proposing Federated Text-driven Prompt Generation (FedTPG), which learns a unified prompt generation network across multiple remote clients in a scalable manner. The prompt generation network is conditioned on task-related text input, thus is context-aware, making it suitable to generalize for both seen and unseen classes. Our comprehensive empirical evaluations on nine diverse image classification datasets show that our method is superior to existing federated prompt learning methods, achieving better overall generalization on both seen and unseen classes, as well as datasets.

## 1 INTRODUCTION

Vision-language models (VLMs) have recently emerged as a transformative technology for machine learning applications. Seminal contributions like Contrastive Language-Image Pretraining (CLIP) Radford et al. (2021) have demonstrated unprecedented capabilities in diverse image classification tasks. One often leverages manually-engineered text prompts, such as "a photo of a [class]," to utilize CLIP's rich semantic features (Jia et al., 2021). CLIP has shown its robustness and versatility in handling a wide range of image distributions. These properties make CLIP naturally aligned with the objective of Federated Learning (FL), a decentralized approach to train machine learning models with data privacy. However, high computational and communication costs associated with server-client interaction make the training of CLIP impractical in the FL setting. This motivates us to explore more efficient and effective methods to adapt the advantages of CLIP in FL.

Emerging prompt learning methodologies based on CLIP such as Context Optimization (CoOp) have revealed that fine-tuning CLIP can be made more efficient by substituting hand-crafted prompts with learnable soft prompt vectors in a few-shot learning paradigm (Perez et al., 2021) for one downstream task in centralized learning (Zhou et al., 2022b;a; Zhu et al., 2022; Yao et al., 2023).Existing federated prompt learning method, Federated Context Optimization (FedCoOp) (Guo et al., 2023b), adapts the learning paradigm of CoOp to FL by learning a unified set of prompt vectors across multiple clients with different datasets. FedCoOp improves over CLIP on the seen (during training) classes in each client, but it struggles to generalize on the unseen classes (not included in training). Similarly, prompt vectors optimized on seen classification tasks fail to generalize to new tasks of

---

[*]Equal contribution. Correspondence to: Chen.Qiu@us.bosch.com
[†]Work done during internship at Bosch Center for AI, USA.

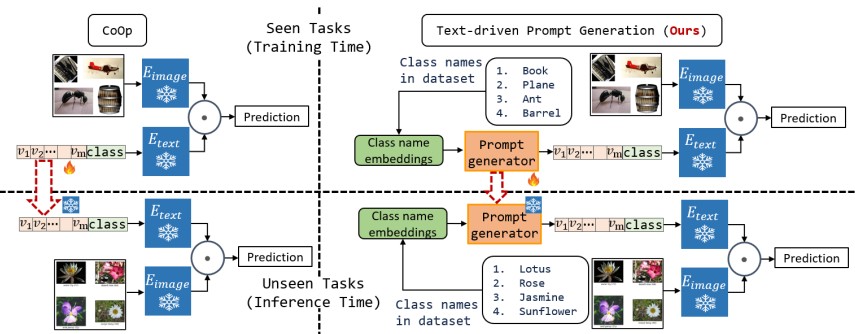

Figure 1: Our proposed prompt generator generates prompt vectors conditioning on the target classification task-related text input. Leveraging contextual awareness, the generated prompt vectors enrich CLIP with context information in the text input and can generalize to unseen classes.

different contexts (e.g., from object recognition to texture classification). Unless otherwise noted, we refer to "task" as an image classification dataset within the context of this work.

Instead of learning one unified set of prompt vectors for different classification tasks, we propose to convert text input containing task-specific semantic information to context-aware prompt vectors. Benefiting from context information in text input, we aim to generate prompt vectors that generalize well to classification tasks that have not been previously observed (refer Figure 1 for an illustration of the concept). Following that, we propose **Fed**erated **T**ext-driven **P**rompt **G**eneration (FedTPG), which learns a lightweight unified prompt generator across multiple clients collaboratively. Each client optimizes the prompt generator locally for its classification task described by few-shot image-text pairs, followed by the FL server-client communication to obtain the global prompt generator model. An overview of our FedTPG with two remote clients is shown in Figure 2. By training on various image classification tasks, our prompt generator learns to generate prompt vectors conditioned on context-related text inputs. Leveraging contextual awareness, the generated prompt vectors differentiate themselves across various tasks and enrich CLIP with context information of the target task. Our comprehensive evaluation on nine diverse image classification datasets demonstrate that FedTPG has improved generalization over the existing prompt learning method FedCoOp on unseen classes by $4.32\%$ and unseen datasets by $1.82\%$, on average.

We summarize the contributions of our work as follows: (1) We develop a text-driven prompt generation (TPG) technique to improve the generalization performance from observed classification tasks to new classification tasks with different contexts. Instead of learning fixed prompt vectors, the prompt generator converts task-related text input to context-aware prompt vectors for various image classification tasks. (2) We propose FedTPG, a scalable way of learning a unified, generalized text-driven prompt generator across multiple clients with various classification tasks collaboratively. (3) We undertake exhaustive empirical analysis using nine datasets to validate the efficacy of FedTPG. Our comparative studies with existing federated prompt learning methods demonstrate FedTPG's superior generalization performance on image classification tasks encompassing a range of domains.

## 2 RELATED WORK

**Visual-Language Model Prompt Learning.** Prompt learning, a variation of fine-tuning VLMs, has shown considerable promise in enhancing the task-specific performance of existing pre-trained models under few-shot settings. A significant advancement in this direction was CoOp (Zhou et al., 2022b), which introduced the notion of optimizing continual prompt context vectors for better task adaptation. CoCoOp (Zhou et al., 2022a) generates prompt vectors conditioning on images with a neural network. Several other works have also explored the interplay between textual prompts and visual inputs (Zang et al., 2022; Li et al., 2023b; Xing et al., 2022). Specifically, MaPLe (Khattak et al., 2023) extends the prompt learning paradigm by integrating both visual and textual information for a more robust task adaptation. Bulat & Tzimiropoulos (2023); Yao et al. (2023) explore text-to-text optimization to encourage natural language-aware soft prompting in VLMs. Chen et al. (2022; 2023b) extend CoOp by learning multiple prompts to describe different characteristics of a category. Concurrent with our work, Wang et al. (2023b); Udandarao et al. (2023) propose to utilize auxiliary images to improve the model performance on unseen tasks. Different than learning one model for

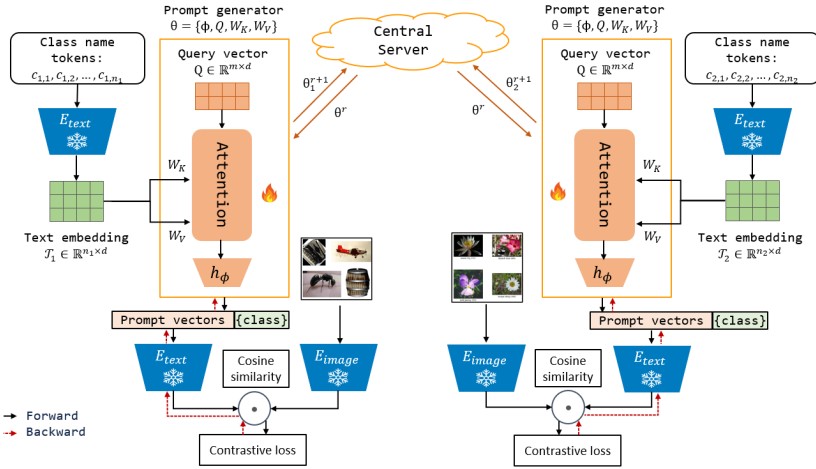

Figure 2: Our proposed FedTPG learns a unified prompt generator over the frozen CLIP model for converting task-related text input $\mathcal{T}$ to context-aware prompt vectors. The prompt generator is learned across multiple clients with different classification datasets collaboratively.

one task in all prior works, we focus on learning one unified model that generalizes well across a broad spectrum of tasks and domains with task-related text inputs.

**Federated Learning with Visual-Language Models.** Federated Learning (FL) (McMahan et al., 2017) has emerged as a pivotal paradigm for decentralized training of machine learning models on heterogeneous data (Li et al., 2023a).Recently, fine-tuning of VLMs has been extended to the federated setting to reduce the computational burden on a single device.FedCLIP (Lu et al., 2023) proposes an extension of standard fine-tuning of CLIP to the FL setting to enable strong performance and personalization. Halbe et al. (2023) provides a continual lifelong prompt learning mechanism to mitigate the effect of client drift. Wang et al. (2023a) further showcase the corrective attribute of prompts in the snapshot compressive imaging application domain while Chen et al. (2023a) highlight the adaptability of federated prompt-based methods to diverse data landscapes beyond visual and textual data for weather forecasting. Of relevance to our approach is PromptFL (Guo et al., 2023b)[1], which proposes a FL framework for prompt learning that enables participants to cooperatively learn a common prompt vector. Su et al. (2022) who delve into the cross-domain applicability of federated prompt learning in VLMs, and Guo et al. (2023a) who combine a federated prompt learning scheme with personalized spacial visual features. A key distinction between these methods and our approach is our use of a learnable text-conditioned prompt generator which improves generalization performance on both seen and unseen tasks, a typically unexplored setting for VLMs under the FL scheme. Concurrent with our work, Yang et al. (2023) propose a prompt generator with a cross-attention mechanism similar to our approach. In addition to their focus on using a frozen ViT backend, we hypothesize that their dependence on fixed client-specific features learned from seen clients would limit their generalization to unseen tasks. In comparison, our prompt generation depends on text inputs and has no hurdles in generalizing to unseen tasks.

## 3 METHOD

In this section, we present our problem setup of FL in Section 3.1, followed by our text-driven prompt generation technique in Section 3.2 and finally propose our FedTPG algorithm that deploys text-driven prompt generation in FL in Section 3.3.

### 3.1 PROBLEM SETUP

We consider a federated network setting with one central server for model aggregation and multiple remote clients, where each client $i$ has a private classification dataset with labeled images $(x, y) \sim \mathcal{D}_i$ from $n_i$ classes with class name tokens $\{c_{i,j}\}_{j=1}^{n_i}$ (a sample setting with two remote clients is depicted in Figure 2). Data distribution across the federated network follows a non-IID setup where clients contain samples from a disjoint set of classes. The goal of FL framework in our setup is

---

[1]For the sake of presentation, we name PromptFL as FedCoOp, as PromptFL adapts CoOp to the FL setting.

to jointly learn one model that not only solves different image classification tasks spanning across multiple remote clients but also attains generalization ability to unseen classes and datasets. In contrast to the setting in FL literature (Kairouz et al., 2021), our consideration of generalization to unseen classes and datasets makes the setup more challenging. Following the recent success of VLMs like CLIP across a broad range of tasks (Radford et al., 2021), we look into the adaptation of CLIP models in our FL framework to achieve our goal of generalization.

CLIP is a large VLM with an image encoder $E_{image}$ and a text encoder $E_{text}$, and can classify images utilizing linguistic knowledge. In our FL setup, we consider each client to have access to an off-the-shelf pretrained CLIP model. We focus on adapting the pretrained CLIP model collaboratively across all clients. However, updating large models like CLIP requires extensive computational power and bandwidth, making it impractical for FL applications. Recently, prompt learning has been used to offer a computation and communication efficient FL framework e.g., FedCoOp (Guo et al., 2023b) for adapting a frozen CLIP across multiple clients. Specifically, hand-crafted text prompts (e.g., "a photo of a [class]") for $E_{text}$ are replaced with trainable prompt vectors $v_1, v_2, ..., v_m$, while keeping CLIP weights unaltered. In federated prompt learning, lightweight trainable prompt vectors are shared across clients at each communication round and updated with local training on client data.

In this work, our goal is to learn a FL prompt model that can solve various image classification tasks across multiple clients and also generalize to novel classes or image classification tasks from new clients, which can be challenging to existing methods like FedCoOp. Zhou et al. (2022a) have shown that CoOp's prompt vectors, optimized for observed classes, fail to generalize to novel classes. We notice a similar generalization issue in FedCoOp i.e., learned unified prompt vectors perform well on the seen classification tasks across remote clients, but fail to generalize to tasks with different contexts (e.g., from object recognition to texture classification). We attribute this behavior to the fixed nature of soft prompts and not being able to adjust to the context of the task. To address this, we propose a novel strategy that alters how the soft prompt vectors are obtained. Instead of directly learning the soft prompts, we learn a text-driven prompt generation module that takes task-related text input and transforms it into context-aware prompt vectors, which we detail in the next section.

## 3.2 TEXT-DRIVEN PROMPT GENERATION

We develop a prompt generation module $f_\theta$ that generates context-aware prompt vectors conditioned on the target classification task-related text inputs, as shown in Figure 1. The text input is translated to text embeddings $\mathcal{T}$ and the prompt generator $f_\theta$ converts these text embeddings $\mathcal{T}$ to a set of $m$-length input prompt vectors $\mathcal{P} \in \mathbb{R}^{m \times d}$ for $E_{text}$ as:

$$\mathcal{P} = \{v_k\}_{k=1}^m = f_\theta(\mathcal{T}). \tag{1}$$

Context-related text input can be obtained from the natural language description. We find that available candidate class names naturally represent context-related text for the classification task ([class 0], [class 1], ..., [class $n$]). We translate the natural language class names to text embeddings as $\mathcal{T} = \{E_{text}(c_j)\}_{j=1}^n \in \mathbb{R}^{n \times d}$, a set of embeddings of $n$ class name tokens $c_j$ from CLIP text encoder[2]. Besides, prompt generator $f_\theta$ is a lightweight cross-attention module comprising of learnable parameters $\phi, Q \in \mathbb{R}^{m \times d}, W_K \in \mathbb{R}^{d \times d}, W_V \in \mathbb{R}^{d \times d}$. Given the text embeddings $\mathcal{T}$ we have:

$$f_\theta(\mathcal{T}) = h_\phi(\text{CrossAttention}(Q, K_\mathcal{T}, V_\mathcal{T})) \quad \text{with} \quad K_\mathcal{T} = \mathcal{T} \times W_K, \quad V_\mathcal{T} = \mathcal{T} \times W_V. \tag{2}$$

The prompt generator transforms context information from the text embeddings $\mathcal{T}$ into key and value vectors $K_\mathcal{T}$ and $V_\mathcal{T}$ respectively. Cross-attention layer merges these vectors with the learnable query vector $Q$, and hidden layers $h_\phi$ projects cross-attention layer output to prompt vectors $\mathcal{P}$.

Prompt vector for each class $j$ is defined as $t_j = \mathcal{P} \cup \{c_j\}$, concatenating generated context prompt vectors $\mathcal{P}$ and text token of class name $c_j$. Given an input image $x$ and prompt vectors for all $n$ candidate classes, the prediction probability of CLIP for a classification task is computed as follows:

$$p_\theta(y = j|x, \mathcal{T}) = \frac{\exp(\cos(E_{image}(x), E_{text}(t_j))/\tau)}{\sum_i^n \exp(\cos(E_{image}(x), E_{text}(t_i))/\tau)}. \tag{3}$$

Text embeddings $\mathcal{T}$ produced from a well-pretrained text encoder like CLIP provides rich and meaningful context information for a given text. The prompt generator $f_\theta$ should serve to extract and

---

[2]For simplicity, we consider a single client with index $i = 1$, and remove the client's index $i$ in notations.

---

**Algorithm 1:** FedTPG Algorithm

---

**Input:** No. of communication rounds $R$, No. of local epochs $K$, initialization parameters $\theta^0$.

**Server executes:**

Initialize prompt generator $f_\theta$ parameters with $\theta^0$.

**for** $r \leftarrow 0$ **to** $R$ **do**

  Pick a random subset of remote clients as $\mathcal{S}^r$.

  **for** $i \in \mathcal{S}^r$ in parallel **do**

   Send the current global model $\theta^r$ to client $i$.

   Receive locally updated $\theta_i^{r+1}$ from **Local Client Training**.

  **end**

  Aggregate the updated model parameters $\theta^{r+1} = \frac{1}{|\mathcal{S}^r|} \sum_{i \in \mathcal{S}^r} \theta_i^{r+1}$.

**end**

Obtain the final model parameter $\theta^R$.

**Local Client Training:**

Obtain the set of class name embeddings $\mathcal{T}_i = \{E_{text}(c_{i,j})\}_{j=1}^{n_i}$.

**for** $k \leftarrow 0$ **to** $K$ **do**

  Generate the context prompt vectors $\mathcal{P}_i^r = f_{\theta_i^r}(\mathcal{T}_i)$.

  Get the prompt vectors for each class $t_{i,j}^r = \mathcal{P}_i^r \cup \{c_{i,j}\}$.

  Update parameters $\theta^r$ to $\theta_i^{r+1}$ locally using eqs. (3) to (5) on $(x, y) \sim \mathcal{D}_i$.

**end**

---

transfer context-critical information from the already meaningful embeddings $\mathcal{T}$ to prompt vectors $\mathcal{P}$. Training $f_\theta$ on different classification tasks from diverse contexts would facilitate its convergence to produce generalized context-aware prompt vectors, and thus improve prediction precision of $p_\theta(y = j|x, \mathcal{T})$ on unseen classes. In practical scenarios, the data encompassing a wide range of classification tasks is typically distributed across different clients. Addressing this, we next present a scalable way of learning the prompt generator across multiple clients collaboratively.

## 3.3 LOCAL TRAINING AND SERVER AGGREGATION

We incorporate our prompt generation module in FL settings, where multiple remote clients handling diverse image classification tasks train the prompt generator $f_\theta$ collaboratively. We refer this approach as **Fed**erated **T**ext-driven **P**rompt **G**eneration (FedTPG). We outline the training pipeline of our FedTPG in Algorithm 1. Initially, the server initializes $f_\theta$ parameters randomly with $\theta^0$ and then at each communication round, a random subset of remote clients $\mathcal{S}^r$ retrieve the up-to-date $f_\theta$ parameters for local training. Below we describe the training steps of FedTPG at each round $r$:

- **Step I:** Remote client $i$ in $\mathcal{S}^r$ receives current up-to-date parameters $\theta^r$ to configure the local $f_{\theta^r}$.

- **Step II:** At each client, the frozen CLIP text encoder $E_{text}$ provides text embeddings of the local available class name tokens $\mathcal{T}_i = \{E_{text}(c_{i,j})\}_{j=1}^{n_i}$. The prompt generator $f_{\theta^r}$, the frozen CLIP model, the context text embeddings $\mathcal{T}_i$, and the dataset $\mathcal{D}_i$ together define the local objective as:

$$L_i(\theta^r; \mathcal{T}_i) = -\mathbb{E}_{(x,y) \in \mathcal{D}_i} y \log p_{\theta^r}(y|x, \mathcal{T}_i), \tag{4}$$

  where $p_{\theta^r}(y|x, \mathcal{T}_i)$ is defined in eq. (3). By utilizing an optimizer, e.g. SGD, we can estimate the unbiased gradient of $L_i(\theta^r; \mathcal{T}_i)$ with respect to $\theta^r$ and get the updated parameters $\theta_i^{r+1}$ after $K$ iterations with a learning rate $\eta^r$ as:

$$\theta_i^{r+1} = \text{SGD}_K(\eta^r, \theta^r, \mathcal{T}_i, L_i) \tag{5}$$

- **Step III:** After local few-shot training, all the remote clients in $\mathcal{S}^r$ send back their locally updated prompt generator $\theta_i^{r+1}$ to the server for aggregation: $\theta^{r+1} = \frac{1}{|\mathcal{S}^r|} \sum_{i \in \mathcal{S}^r} \theta_i^{r+1}$.

After performing Step I-III for $R$ communication rounds, FedTPG obtains the final model parameters $\theta^R$. We argue that the proposed FedTPG can achieve the generalization goal from two aspects: (1) unlike existing prompt learning techniques that directly learn a fixed prompt vector, our TPG method captures a richer contextual and semantic information for each local classification task; (2)

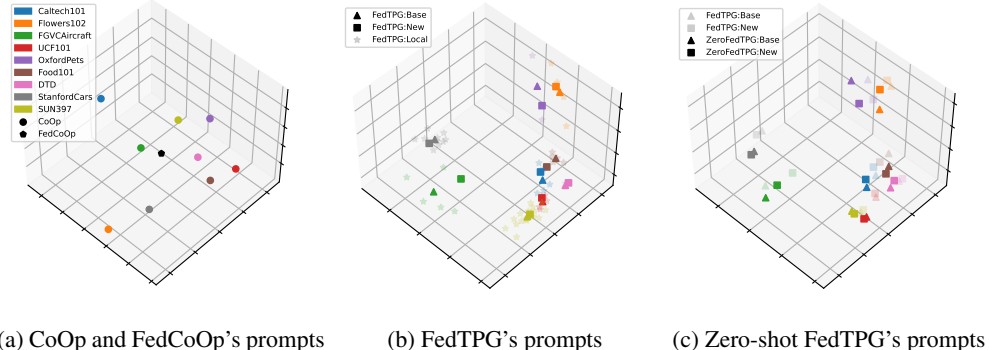

(a) CoOp and FedCoOp's prompts     (b) FedTPG's prompts     (c) Zero-shot FedTPG's prompts

Figure 3: 3D visualization (after PCA) of soft prompt vectors. (a) CoOp learns diverged prompt vectors on each dataset individually, while FedCoOp learns one unified set of prompt vectors for tasks with various contexts (b) FedTPG's prompt generator learned on bases classes generates context-aware prompt vectors for each task. (c) FedTPG's prompt generator learned on ImageNet generates context-ware prompt vectors for nine unseen datasets aligned with the generated vectors in (b).

through the FL collaboration framework, diverse contextual and semantic information across multiple remote clients with different tasks benefit the model learning well. Multiple clients encode text embeddings based on their distinct tasks, enabling the global model to serve a variety of contexts without overfitting to a specific task. Overall, the federated model can potentially learn a richer set of semantic features, and facilities better "transfer learning" capabilities, enabling the model to generalize well to both seen and new unseen tasks (that includes both classes and datasets).

## 4 EXPERIMENTS

We evaluate the proposed method FedTPG mainly on two benchmarks: (1) generalization to unseen related *classes* in Section 4.1, (2) generalization to unseen *datasets* in Section 4.2. We also provide ablation studies to evaluate FedTPG's robustness under various settings in Section 4.3.

**Baselines.** We compare FedTPG with (i) CLIP with hand-crafted text prompt template, e.g., "a photo of a [class]"; (ii) CoOp (Zhou et al., 2022b) with learnable prompt vectors replacing hand-crafted text prompts. CoOp is trained on each client individually to provide a baseline of local training. (iii) FedCoOp (Guo et al., 2023b), a FL variant of CoOp. The unified prompt vectors are learned across multiple clients with federated averaging. (iv) FedKgCoOp, a FL variant of KgCoOp (Yao et al., 2023). KgCoOp improves over CoOp on generalization performance by adding a regularization of minimizing the discrepancy between learned prompts and the hand-crafted prompts. (v) FedCoCoOp, a FL variant of CoCoOp (Zhou et al., 2022a). CoCoOp generates prompt vectors conditioning on images with a neural network. (vi) FedMaple, a FL variant of Maple (Khattak et al., 2023). Maple learns prompt vectors for both vision and text encoders. We develop all FL variants of existing prompt learning approaches with FedAvg (McMahan et al., 2017). For all FL methods, one unified model learned across clients is used for the evaluation of different datasets.

**Implementation Details.** All methods are built on a frozen CLIP with ViT-B/16 backbone. FedTPG learns a unified prompt generator parameterized by a four-head cross-attention layer with layer norm and a MLP ($h_\phi$) consisting of two linear layers with ReLU. The dimension of vectors $Q$, $K_\mathcal{T}$, $V_\mathcal{T}$ in the cross-attention layer, and linear layers in $h_\phi$ is 512. The length $m$ of prompt vectors is 4, and the dimension $d$ is 512. Please refer to Appendix A and Appendix B for more details.

### 4.1 GENERALIZATION TO SEEN AND UNSEEN CLASSES

**Datasets.** We employ nine image datasets including Caltech101 (Fei-Fei et al., 2004), OxfordPets (Parkhi et al., 2012), StanfordCars (Krause et al., 2013), Flowers102 (Nilsback & Zisserman, 2008), Food101 (Bossard et al., 2014), FGVCAircraft (Maji et al., 2013), SUN397 (Xiao et al., 2010), UCF101 (Soomro et al., 2012), and DTD (Cimpoi et al., 2014).

**Experimental setup.** We split the classes of each dataset equally into two groups, one as base classes and the other as new classes. Images from base classes are available for training, while the images from new classes are used for evaluating the generalization performance. We perform all

Table 1: Accuracies (%) on clients' local tasks (seen), base (seen) classes, and new (unseen) classes. FedTPG achieves the superior generalization performance over existing prompt learning methods and their FL variants, and the highest harmonic mean (HM) of three benchmark results.

(a) Average over 9 datasets.

|  | Local | Base | New | HM |
|---|---|---|---|---|
| CLIP | 76.72 | 70.52 | 75.78 | 74.24 |
| CoOp | **83.67** | 71.49 | 71.15 | 75.01 |
| FedCoOp | 81.75 | **74.50** | 71.70 | 75.75 |
| FedKgCoOp | 78.38 | 72.18 | 75.87 | 75.39 |
| FedCoCoOp | 81.41 | 73.87 | 65.98 | 73.21 |
| FedMaple | 81.63 | 74.44 | 70.59 | 75.28 |
| FedTPG | 80.75 | 73.68 | **76.02** | **76.70** |

(b) Caltech101.

|  | Local | Base | New | HM |
|---|---|---|---|---|
| CLIP | 97.57 | 96.97 | 93.89 | 96.12 |
| CoOp | **97.79** | 94.02 | 93.1 | 94.93 |
| FedCoOp | 96.97 | 96.69 | 92.79 | 95.45 |
| FedKgCoOp | 97.65 | **97.24** | 94.79 | 96.54 |
| FedCoCoOp | 96.71 | 94.41 | 91.59 | 94.19 |
| FedMaple | 97.00 | 95.41 | 90.06 | 94.06 |
| FedTPG | 97.59 | 97.08 | **95.24** | **96.62** |

(c) Flowers102.

|  | Local | Base | New | HM |
|---|---|---|---|---|
| CLIP | 82.58 | 72.18 | **77.94** | 77.33 |
| CoOp | **97.27** | 69.37 | 71.95 | 77.73 |
| FedCoOp | 94.44 | **76.40** | 70.12 | 79.16 |
| FedKgCoOp | 84.59 | 72.11 | 77.06 | 77.59 |
| FedCoCoOp | 94.00 | 77.49 | 65.63 | 77.31 |
| FedMaple | 94.28 | 76.44 | 68.51 | 78.35 |
| FedTPG | 90.76 | 71.80 | 77.76 | **79.35** |

(d) FGVCAircraft.

|  | Local | Base | New | HM |
|---|---|---|---|---|
| CLIP | 30.59 | 27.55 | **35.81** | 30.96 |
| CoOp | **36.88** | 28.30 | 28.59 | 30.79 |
| FedCoOp | 36.29 | **32.41** | 30.95 | 33.07 |
| FedKgCoOp | 33.68 | 29.79 | 34.01 | 32.37 |
| FedCoCoOp | 35.21 | 31.93 | 22.67 | 28.89 |
| FedMaple | 35.83 | 31.39 | 32.34 | 33.08 |
| FedTPG | 34.68 | 30.82 | 35.18 | **33.44** |

(e) UCF101.

|  | Local | Base | New | HM |
|---|---|---|---|---|
| CLIP | 80.75 | 70.58 | **77.5** | 76.04 |
| CoOp | **88.37** | 69.62 | 68.09 | 74.32 |
| FedCoOp | 86.13 | **75.65** | 70.60 | 76.93 |
| FedKgCoOp | 82.66 | 73.14 | 76.36 | 77.19 |
| FedCoCoOp | 84.92 | 75.23 | 64.25 | 73.83 |
| FedMaple | 84.17 | 75.12 | 68.68 | 75.46 |
| FedTPG | 85.64 | 74.89 | 76.64 | **78.79** |

(f) OxfordPets.

|  | Local | Base | New | HM |
|---|---|---|---|---|
| CLIP | 91.33 | 91.33 | 97.04 | 93.16 |
| CoOp | 94.89 | 94.89 | 96.60 | 95.46 |
| FedCoOp | 93.31 | 93.32 | 95.39 | 94.00 |
| FedKgCoOp | 91.58 | 91.58 | 96.53 | 93.17 |
| FedCoCoOp | 92.34 | 92.34 | 87.36 | 90.61 |
| FedMaple | **95.00** | **95.00** | **97.09** | **95.68** |
| FedTPG | 94.70 | 94.69 | 95.79 | 95.06 |

(g) Food101.

|  | Local | Base | New | HM |
|---|---|---|---|---|
| CLIP | **94.39** | 90.16 | 91.25 | 91.90 |
| CoOp | 93.98 | 88.20 | 88.22 | 90.05 |
| FedCoOp | 93.52 | 88.63 | 88.47 | 90.15 |
| FedKgCoOp | 94.19 | **89.94** | **91.81** | **91.95** |
| FedCoCoOp | 93.24 | 87.57 | 84.95 | 88.45 |
| FedMaple | 93.95 | 89.43 | 89.60 | 90.94 |
| FedTPG | 94.09 | 89.87 | 91.64 | 91.83 |

(h) DTD.

|  | Local | Base | New | HM |
|---|---|---|---|---|
| CLIP | 53.13 | 53.01 | 58.21 | 54.68 |
| CoOp | **72.34** | **72.34** | 54.99 | **65.46** |
| FedCoOp | 68.67 | 68.67 | 52.74 | 62.39 |
| FedKgCoOp | 58.76 | 58.75 | 59.61 | 59.04 |
| FedCoCoOp | 68.63 | 68.63 | 45.77 | 58.83 |
| FedMaple | 68.28 | 68.28 | 46.61 | 59.11 |
| FedTPG | 63.62 | 63.62 | **60.51** | 62.55 |

(i) StanfordCars.

|  | Local | Base | New | HM |
|---|---|---|---|---|
| CLIP | 71.51 | 63.44 | 74.9 | 69.61 |
| CoOp | **78.65** | 61.34 | 70.17 | 69.33 |
| FedCoOp | 74.53 | 66.16 | 72.32 | 70.82 |
| FedKgCoOp | 71.89 | 64.33 | **75.71** | 70.32 |
| FedCoCoOp | 76.62 | 66.51 | 66.40 | 69.52 |
| FedMaple | 74.76 | 66.26 | 71.33 | 70.60 |
| FedTPG | 74.54 | **66.34** | 74.26 | **71.50** |

(j) SUN397.

|  | Local | Base | New | HM |
|---|---|---|---|---|
| CLIP | 88.66 | 69.41 | 75.46 | 77.05 |
| CoOp | **92.83** | 65.29 | 68.62 | 73.78 |
| FedCoOp | 91.93 | 72.34 | 71.89 | 77.70 |
| FedKgCoOp | 90.38 | 72.72 | 76.94 | 79.34 |
| FedCoCoOp | 91.44 | 69.76 | 65.36 | 73.94 |
| FedMaple | 91.40 | 72.66 | 71.33 | 77.47 |
| FedTPG | 91.11 | **74.01** | **77.13** | **80.10** |

FL methods under a non-IID FL setting, where the base classes of all nine datasets are distributed to multiple clients. Each client owns $n = 20$ completely disjoint classes where each class has eight labeled images for few-shot training. We report the classification accuracies on clients' local classification tasks, on the base classes (combining classes from multiple clients), on the new

Table 2: Accuracies (%) on ImageNet (seen) and domain-shifted ImageNet variants (unseen). FedTPG consistently outperforms other baselines on both source dataset and domain-shifed datsets.

| | ImageNet | ImageNetV2 | ImageNet-S | ImageNet-A | ImageNet-R | Average |
|---|---|---|---|---|---|---|
| CLIP | 66.75 | 60.79 | 46.12 | 47.79 | 74.00 | 57.18 |
| FedCoOp | 67.80 | 61.59 | 45.61 | 48.78 | 74.49 | 57.62 |
| FedKgCoOp | 67.53 | 61.60 | 46.69 | 48.37 | 74.71 | 57.84 |
| FedCoCoOp | 68.51 | 62.29 | 46.90 | **50.33** | **76.49** | 59.00 |
| FedMaple | 66.96 | 60.65 | 44.69 | 46.24 | 74.62 | 56.55 |
| FedTPG | **69.51** | **62.90** | **47.65** | 49.97 | 76.35 | **59.22** |

Table 3: Accuracies (%) on source (seen) and target (unseen) datasets. FedTPG consistently outperforms other federated prompt learning methods on both source dataset and unseen target datsets.

| | Source | Target | | | | | | | | | | |
|---|---|---|---|---|---|---|---|---|---|---|---|---|
| | ImageNet | Caltech101 | Flowers102 | FGVCAircraft | UCF101 | OxfordPets | Food101 | DTD | StanfordCars | SUN397 | EuroSAT | Average |
| CLIP | 66.75 | 92.90 | **71.29** | **24.72** | 66.75 | 89.15 | **86.09** | 44.33 | **65.29** | 62.59 | 47.68 | 65.08 |
| FedCoOp | 67.80 | 91.87 | 68.13 | 21.44 | 64.13 | 88.70 | 85.85 | 42.43 | 63.59 | 62.77 | 43.26 | 63.22 |
| FedKgCoOp | 67.53 | 93.63 | 69.31 | 23.06 | 64.46 | 88.55 | 85.37 | 44.74 | 64.99 | 63.85 | 43.29 | 64.13 |
| FedCoCoOp | 68.51 | **94.11** | 66.34 | 20.79 | 62.75 | 89.04 | 85.40 | 43.20 | 63.98 | 64.02 | **55.40** | 64.50 |
| FedMaple | 66.96 | 92.49 | 68.25 | 23.52 | 60.32 | 89.67 | 83.52 | 44.68 | 60.16 | 61.85 | 45.38 | 62.98 |
| FedTPG | **69.51** | 93.75 | 70.04 | 23.22 | **64.72** | **90.60** | 85.91 | **46.25** | 63.98 | **66.78** | 47.86 | **65.31** |

classes in Table 1. We report the harmonic mean (HM) of these three accuracies showing the overall performance. All results are averaged over three independent runs.

**Quantitative results.** As shown in Table 1(a), the proposed FedTPG achieves the best average accuracy on new classes, showing its advanced generalization ability. FedTPG also achieves the highest harmonic mean which averages the accuracies on clients' local tasks, base classes, and new classes. Although the prompt generator is trained on local tasks consisting of a few classes, it generalizes well to a more complex classification task one the base classes (combining classes from multiple clients), and a novel classification task on the unseen classes. Due to the non-IID setting, CoOp outperforms the FL methods on the corresponding local task but fails to generalize to other base classes and new classes. Benefiting from learning across multiple clients, FedCoOp, FedCoCoOp and FedMalpe improve a lot on base classes, however, have degraded performance on new classes, highlighting the generalization challenge in federated prompt learning. FedKgCoOp has an improved accuracy on new classes with a cost of performance degradation on base classes and local tasks, resulting from the difficulties of balancing the CLIP loss and the regularization term.

**Qualitative analysis.** We visualize the prompt vectors learned by CoOp on each dataset individually and the unified prompt vectors learned by FedCoOp in Figure 3 (a), and the prompt vectors generated by FedTPG in Figure 3 (b). We can see that CoOp learns different optimal prompt vectors on each dataset. However, the unified prompt vectors learned by FedCoOp are not flexible enough to fit the context of all different datasets. In comparison, FedTPG learns to generate task-specific prompt vectors conditioning on the context-related text input. From Figure 3 (b) we can see the prompt vectors generated on clients (stars) sharing data from the same dataset are automatically clustered together, showcasing that the prompt generator learns to extract context information from the text input. Also, although the model is not trained on base-class classification and new-class classification, their associated generated prompt vectors (triangle for base, square for new) are clustered based on the dataset context accordingly, explaining FedTPG's strong generalization ability.

## 4.2 Generalization to unseen datasets

**Datasets.** For evaluating the generalization to unseen datasets, we train all models on ImageNet, and test the model on two benchmarks: (1) four variants of ImageNet containing various types of domain shifting: including ImageNetV2, ImageNet-Sketch, ImageNet-A, and ImageNet-R; (2) ten unseen datasets including nine datasets used in Table 1 and EuroSAT (Helber et al., 2019).

Table 4: Ablation study: three trials where each client owns $n = \{5, 10, 20\}$ disjoint classes. FedTPG consistently achieves the highest harmonic mean (HM).

| | n=5 | | | n=10 | | | n=20 | | |
| --- | --- | --- | --- | --- | --- | --- | --- | --- | --- |
| | Base | New | HM | Base | New | HM | Base | New | HM |
| FedCoOp | 69.53 | 70.05 | 69.69 | **72.15** | 70.61 | 71.37 | **74.50** | 71.70 | 73.07 |
| FedKgCoOp | 70.83 | **75.55** | 73.11 | 71.18 | 75.81 | 73.42 | 72.18 | 75.87 | 73.98 |
| FedTPG | **71.08** | 75.51 | **73.23** | 72.15 | **75.84** | **73.95** | 73.68 | **76.02** | **74.83** |

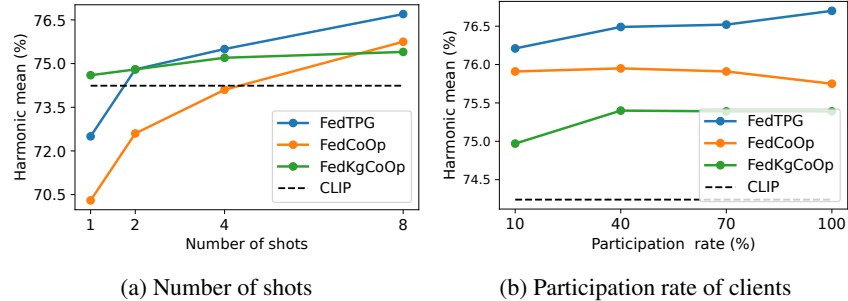

(a) Number of shots                 (b) Participation rate of clients

Figure 4: (a) FedTPG gets improved when increasing the number of shots (for training), and has the best results when using more than one shot. (b) FedTPG is robust to the participation rate of clients.

**Experimental setup.** We consider a non-IID setting with 200 clients. Each client owns $n = 5$ disjoint classes. At each communication round, random $10\%$ clients contribute to the model update. We also consider a few-shot setting, where eight labeled images are available in each class. We only consider FL baselines in this setting. All results are averaged over three independent runs.

**Results.** Our proposed FedTPG improves over other FL prompt methods on ImageNet and other variants of ImageNet consistently as shown in Table 2. On the more challenging unseen datasets, while all the existing methods sacrifice their performance, FedTPG avoids the overfitting problem as the compared FL prompt methods, outperforming CLIP as shown in Table 3. Although the prompt generator is trained on images and class names from ImageNet, the model learns a generalizable function mapping the context-related text embeddings $\mathcal{T}$ to task-specific prompt vectors as visualized in Figure 3 (c), improving the classification accuracy on datasets with totally different context, e.g., from object recognition to texture classification. We can see that the prompt vectors generated by FedTPG trained on ImageNet are aligned with the prompt vectors generated by FedTPG trained on these nine datasets, which demonstrates its cross-dataset transferability.

### 4.3 ABLATION STUDIES

**Size of clients:** To understand the impact of the number of classes owned by the client, we conduct three trials where each client owns $n = \{5, 10, 20\}$ disjoint classes, and number of shots is 8. As shown in Table 4, FedTPG outperforms FL baselines in all cases in terms of the harmonic mean.

**Number of shots:** In Figure 4(a), we show FedTPG consistently improves over FedCoOp with varying number of shots, and outperforms FedKgCoOp, when the number of shots is larger than one.

**Participation rate of clients:** In Figure 4(b), we show that FedTPG consistently outperforms FedCoOp and FedKgCoOp, when the participation rate of clients varies from $10\%$ to $100\%$.

### 5 CONCLUSION

This paper addresses the challenge of generalization in adapting CLIP to the FL setting. We propose a novel Federated Text-driven Prompt Generation (FedTPG) algorithm, which learns a unified prompt generator across multiple clients. The prompt generator converts task-related text inputs to context-aware prompt vectors. Our comprehensive experiments demonstrate FedTPG's superior generalization performance, outperforming existing FL prompt methods by decent margins.

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

## A EXPERIMENT SETUP DETAILS

### A.1 DATASET AND HYPER-PARAMETER DETAILS

We follow the settings in Zhou et al. (2022b) to conduct the experiments in this paper with the nine classification datasets for generalization on seen to unseen classes, and four variants of ImageNet datasets for domain shifting, where the statistical details are presented in Table 5.

For each compared FL approach and each classification task, via grid search, the learning rate of the SGD optimizer was set to $\eta = 0.003$ with a decay rate $1e-5$ and a momentum of $0.9$. The local SGD training step is set to $K = 1$. The number of communication rounds is 500. The batch size is 200. By default, all the experimental results in the paper are obtained by averaging from three runs with different random seeds.

### A.2 FEDERATED LEARNING SETUP DETAILS

**Experimental Setup for Seen and Unseen Classes in Table 1** To evaluate the generalization ability for the proposed FedTPG and compared FL approaches from in the paper, we monitor the model performance on the following three benchmark accuracies: (1) The local classification accuracy, representing the performance of local clients' classification tasks on local available classes; (2) The base classification accuracy, representing the performance against all seen classes (combining classes from multiple clients) in a dataset in the FL network; (3) The new classification accuracy, which indicates the performance on unseen classes but within the domain of seen classes. We report the harmonic mean (HM) of these three accuracies on each classification task, as shown in Table 1.

In the FL data partition process for Table 1, we first split the classes of the considered 9 classification datasets equally into two groups $\mathcal{D}^s$ and $\mathcal{D}^u$, denotes seen and unseen groups respectively. Then we split the classes within $\mathcal{D}^s$ to the 30 remote clients, where each remote client has $n = 20$ classes in each local dataset $\mathcal{D}_i$. For each class, the number of image-text paired data shots is set to 8. During the FL training process, the participation rate of remote clients is set to $100\%$ and the communication round is set to $500$.

**Experimental Setup for Unseen Datasets in Table 2 and Table 3** To evaluate the generalization ability of FedTPG on unseen datasets during training, we consider the following two settings: (1) Domain Shifting, where we monitor the performance of model by training with ImageNet and testing on four variants of ImageNet, including ImageNetV2, ImageNet-Sketch, ImageNet-A, and ImageNet-R; (2) Unseen Datasets, where we evaluate the performance of trained model in (1) on nine unseen datasets, including Caltech101, OxfordPets, StanfordCars, Flowers102, Food101, FGV-CAircraft, SUN397, UCF101, and DTD. During the training process, we set the FL network with 200 remote clients where each client has $n = 5$ classes of 8-shots training data disjointly. The participation rate of remote clients is set to $10\%$ that $|\mathcal{S}^r| = 20$ and the global communication round is set to $R = 500$ to obtain $\theta^R$.

**Experimental Setup for Ablation Study in Table 4 and Figure 4** We study the impact of the number of classes owned by each client at Table 4 from the introduced local, base and new classification accuracies with the same setup in Table 1 where a full client participation is performed with $R = 500$ and number of shots is 8. Specifically, we perform the data partition with the disjoint rule during class splitting: when $n = 5$, we set the number of clients to 119; when $n = 10$, we set the number of clients to 59; and when $n = 20$, we set the number of clients to 20, respectively.

The study of the number of shots is shown in Figure 4(b), where we set the number of clients to 30 with $n = 20$ and the client participation rate is $100\%$ in each round where $R = 500$. The study of the participation rate is shown in Figure 4(b), where we set the number of clients to 30 with $n = 20$ and the number of shots is 8.

Then, we monitor the impact of the FL client participation rate in each communication round as shown in Figure 4(a). We formulate the FL network with 30 clients where $n = 20$ and the number of shots is 8. Four client participation rates in $\{10\%, 40\%, 70\%, 100\}\%$ are considered during the model training process with $R = 500$.

Table 5: Dataset statistical details on class, training and test splits, prompt template.

| Dataset | Classes | Train | Test | Hand-crafted prompt template |
|---|---|---|---|---|
| ImageNet | 1000 | 1.28M | 50,000 | A photo of a **[class]** |
| Caltech101 | 101 | 4,128 | 2,465 | A photo of a **[class]** |
| Flowers102 | 102 | 4,093 | 2,463 | A photo of a **[class]**, a type of flower |
| FGVCAircraft | 100 | 3,334 | 3,333 | A photo of a **[class]**, a type of aircraft |
| UCF101 | 101 | 7,639 | 3,783 | A photo of a person doing **[class]** |
| OxfordPets | 37 | 2,944 | 3,369 | A photo of a **[class]**, a type of pet |
| Food101 | 101 | 50,500 | 30,300 | A photo of a **[class]**, a type of food |
| DTD | 47 | 2,820 | 1,692 | A photo of a **[class]**, a type of texture |
| StanfordCars | 196 | 6,509 | 8,041 | A photo of a **[class]** |
| SUN397 | 397 | 15,880 | 19,850 | A photo of a **[class]** |
| EuroSAT | 10 | 13500 | 8,100 | A centered satellite photo of **[class]** |
| ImageNetV2 | 1000 | N/A | 10,000 | A photo of a **[class]** |
| ImageNet-Sketch | 1000 | N/A | 50,889 | A photo of a **[class]** |
| ImageNet-A | 200 | N/A | 7500 | A photo of a **[class]** |
| ImageNet-R | 200 | N/A | 30,000 | A photo of a **[class]** |

# B ADDITIONAL RESULTS

**Size of clients.** Table 6 and Table 7 show the detailed results of FedTPG and the compared FL baselines on the benchmark of seen and unseen classes with $n = 5$ and $n = 10$, respectively. The results of Table 6 and Table 7 are the detailed results of Table 4 in the main paper, where we would like to claim that the HM results in the main paper are the harmonic mean of the base accuracy and the new accuracy, while the results in Table 6 and Table 7 are the harmonic mean of the local accuracy, the base accuracy and the new accuracy that leads to the difference in some columns. The results show that similar to the results of $n = 20$ in Table 1, the proposed FedTPG achieves the best average accuracy on unseen classes, and achieves the best new performance for 3 tasks while the second best new performance for most of the other tasks. We can also observe that as $n$ increases, the advantage of FedTPG against other approaches becomes more significant. This supports our theoretical claim that the unified prompt generator in FedTPG generalizes better on unobserved classification tasks, especially for challenging scenarios.

Table 6: Accuracies (%) on clients' local tasks (seen), base (seen) classes, and new (unseen) classes. Each client has labeled images from five disjoint classes. The number of shot is 8 and $n = 5$.

(a) Average over 9 datasets.

|  | Local | Base | New | HM |
|---|---|---|---|---|
| CLIP | 86.25 | 70.52 | 75.78 | 76.98 |
| FedCoOp | 89.38 | 69.53 | 70.05 | 74.74 |
| FedKgCoOp | 86.63 | 70.83 | 75.55 | 77.12 |
| FedTPG | 87.78 | 71.08 | 75.51 | **77.51** |

(b) Caltech101.

|  | Local | Base | New | HM |
|---|---|---|---|---|
| CLIP | 97.40 | 96.97 | 93.89 | 96.06 |
| FedCoOp | 97.19 | 93.67 | 92.14 | 94.28 |
| FedKgCoOp | 97.95 | 96.57 | 94.21 | **96.22** |
| FedTPG | 97.31 | 94.00 | 94.43 | 95.22 |

(c) Flowers102.

|  | Local | Base | New | HM |
|---|---|---|---|---|
| CLIP | 91.12 | 72.18 | 77.94 | **79.66** |
| FedCoOp | 97.89 | 70.65 | 74.47 | 79.37 |
| FedKgCoOp | 89.96 | 70.27 | 76.51 | 78.09 |
| FedTPG | 94.20 | 70.23 | 76.77 | 79.20 |

(d) FGVCAircraft.

|  | Local | Base | New | HM |
|---|---|---|---|---|
| CLIP | 49.04 | 27.55 | 35.81 | 35.45 |
| FedCoOp | 55.82 | 25.45 | 26.57 | 31.63 |
| FedKgCoOp | 51.98 | 28.89 | 33.75 | **35.93** |
| FedTPG | 53.62 | 26.38 | 33.92 | 34.87 |

(e) UCF101.

|  | Local | Base | New | HM |
|---|---|---|---|---|
| CLIP | 88.78 | 70.58 | 77.50 | **78.25** |
| FedCoOp | 90.71 | 69.75 | 65.33 | 73.77 |
| FedKgCoOp | 87.68 | 70.06 | 76.14 | 77.29 |
| FedTPG | 88.53 | 71.20 | 75.96 | 77.91 |

(f) OxfordPets.

|  | Local | Base | New | HM |
|---|---|---|---|---|
| CLIP | 96.75 | 91.33 | 97.04 | 94.96 |
| FedCoOp | 98.08 | 91.92 | 94.57 | 94.79 |
| FedKgCoOp | 96.65 | 91.34 | 96.16 | 94.66 |
| FedTPG | 97.96 | 91.39 | 96.03 | **95.04** |

(g) Foods102.

|  | Local | Base | New | HM |
|---|---|---|---|---|
| CLIP | 97.57 | 90.16 | 91.25 | **92.88** |
| FedCoOp | 97.17 | 88.27 | 86.67 | 90.48 |
| FedKgCoOp | 97.42 | 89.59 | 91.52 | 92.72 |
| FedTPG | 97.34 | 89.24 | 91.31 | 92.51 |

(h) DTD.

|  | Local | Base | New | HM |
|---|---|---|---|---|
| CLIP | 79.55 | 53.01 | 58.21 | 61.71 |
| FedCoOp | 86.94 | 54.40 | 51.45 | 60.83 |
| FedKgCoOp | 80.50 | 55.47 | 60.26 | 63.77 |
| FedTPG | 82.72 | 60.19 | 61.53 | **66.73** |

(i) StanfordCars.

|  | Local | Base | New | HM |
|---|---|---|---|---|
| CLIP | 83.06 | 63.44 | 74.90 | 72.90 |
| FedCoOp | 86.06 | 64.84 | 71.77 | 73.22 |
| FedKgCoOp | 83.42 | 63.84 | 75.85 | **73.46** |
| FedTPG | 83.75 | 63.92 | 72.35 | 72.45 |

(j) SUN397.

|  | Local | Base | New | HM |
|---|---|---|---|---|
| CLIP | 93.02 | 69.41 | 75.46 | 78.10 |
| FedCoOp | 94.55 | 66.83 | 67.44 | 74.32 |
| FedKgCoOp | 94.12 | 71.45 | 75.52 | 79.23 |
| FedTPG | 94.56 | 73.17 | 77.24 | **80.67** |

Table 7: Accuracies (%) on clients' local tasks (seen), base (seen) classes, and new (unseen) classes. Each client has labeled images from ten disjoint classes. The number of shot is 8 and $n = 10$.

(a) Average over 9 datasets.

|  | Local | Base | New | HM |
|---|---|---|---|---|
| CLIP | 80.57 | 70.52 | 75.78 | 75.40 |
| FedCoOp | 85.64 | 72.15 | 70.61 | 75.57 |
| FedKgCoOp | 81.39 | 71.18 | 75.81 | 75.90 |
| FedTPG | 83.49 | 72.17 | 75.84 | **76.89** |

(b) Caltech101.

|  | Local | Base | New | HM |
|---|---|---|---|---|
| CLIP | 97.83 | 96.97 | 93.89 | **96.20** |
| FedCoOp | 97.45 | 94.56 | 93.46 | 95.13 |
| FedKgCoOp | 97.64 | 96.80 | 93.99 | 96.12 |
| FedTPG | 98.03 | 95.83 | 94.58 | 96.13 |

(c) Flowers102.

|  | Local | Base | New | HM |
|---|---|---|---|---|
| CLIP | 84.58 | 72.18 | 77.94 | 77.91 |
| FedCoOp | 97.17 | 73.33 | 71.10 | **78.96** |
| FedKgCoOp | 84.77 | 71.93 | 76.80 | 77.48 |
| FedTPG | 90.03 | 71.58 | 77.08 | 78.85 |

(d) FGVCAircraft.

|  | Local | Base | New | HM |
|---|---|---|---|---|
| CLIP | 37.88 | 27.55 | 35.81 | 33.10 |
| FedCoOp | 44.00 | 27.23 | 25.76 | 30.53 |
| FedKgCoOp | 38.53 | 26.86 | 35.06 | 32.71 |
| FedTPG | 41.74 | 28.44 | 35.05 | **34.21** |

(e) UCF101.

|  | Local | Base | New | HM |
|---|---|---|---|---|
| CLIP | 83.65 | 70.58 | 77.5 | 76.87 |
| FedCoOp | 87.56 | 73.53 | 71.76 | 77.01 |
| FedKgCoOp | 84.00 | 71.25 | 76.11 | 76.77 |
| FedTPG | 85.78 | 72.15 | 76.05 | **77.59** |

(f) OxfordPets.

|  | Local | Base | New | HM |
|---|---|---|---|---|
| CLIP | 93.26 | 91.33 | 97.04 | 93.82 |
| FedCoOp | 95.95 | 92.36 | 91.60 | 93.27 |
| FedKgCoOp | 92.55 | 90.32 | 96.36 | 93.01 |
| FedTPG | 95.86 | 93.92 | 96.73 | **95.48** |

(g) Foods102.

|  | Local | Base | New | HM |
|---|---|---|---|---|
| CLIP | 95.94 | 90.16 | 91.25 | **92.38** |
| FedCoOp | 95.18 | 88.21 | 89.91 | 90.72 |
| FedKgCoOp | 95.81 | 89.88 | 91.66 | **92.38** |
| FedTPG | 95.73 | 89.93 | 91.63 | 92.36 |

(h) DTD.

|  | Local | Base | New | HM |
|---|---|---|---|---|
| CLIP | 62.74 | 53.01 | 58.21 | 57.71 |
| FedCoOp | 78.15 | 63.11 | 49.65 | 61.50 |
| FedKgCoOp | 68.10 | 57.12 | 60.26 | 61.49 |
| FedTPG | 71.41 | 59.52 | 60.18 | **63.26** |

(i) StanfordCars.

|  | Local | Base | New | HM |
|---|---|---|---|---|
| CLIP | 78.29 | 63.44 | 74.9 | 71.62 |
| FedCoOp | 81.23 | 65.76 | 70.93 | 72.09 |
| FedKgCoOp | 78.82 | 64.13 | 75.52 | 72.25 |
| FedTPG | 80.15 | 65.33 | 74.62 | **72.84** |

(j) SUN397.

|  | Local | Base | New | HM |
|---|---|---|---|---|
| CLIP | 90.96 | 69.41 | 75.46 | 77.61 |
| FedCoOp | 94.07 | 71.32 | 72.10 | 77.88 |
| FedKgCoOp | 92.28 | 72.36 | 76.47 | 79.51 |
| FedTPG | 92.71 | 72.90 | 76.62 | **79.88** |

