# OpenReview forum: "Federated Text-driven Prompt Generation for Vision-Language Models"
_ICLR.cc/2024/Conference — ICLR 2024 poster_

### Official Review · Reviewer_RTLu · 2023-10-22

**Soundness:** 2 fair
**Presentation:** 2 fair
**Contribution:** 2 fair
**Rating:** 5
**Confidence:** 5

**Summary:**

This paper proposed a Federated Text-driven Prompt Generation (FedTPG) network to generalize the classification ability of vision-language models to unseen classes. The prompt generation network is conditioned on task-related text input, which makes it suitable to generalize for both seen and unseen classes. The experimental results on 9 datasets demonstrate that the proposed FedTPG achieves overall better generalization on both seen and unseen classes.

**Strengths:**

The structure and the content of this paper are complete.

The formulation is clear enough.

The proposed FedTPG is easy to follow.

**Weaknesses:**

1. The applicable scenario of the proposed method is unclear. What scenarios require training vision-language models under the framework of federated learning? And test on unknown datasets?

2. Many existing works have studied generative prompt learning methods [1] [2] [3] and cross-modal prompt learning methods [4] [5] [6]. What is the difference between the proposed FedTPG and these methods? Why the proposed FedTPG can generalize to unknown datasets but the existing methods mentioned above cannot?

3. More comparison experiments with existing prompt learning methods [1-6] should be included to verify the effectiveness of the proposed FedTPG.

4. In Table I, the proposed FedTPG did not achieve the best performance on 9 datasets while under Local Base and HM settings. Does this mean that FedTPG sacrifices the classification performance of known categories when considering unknown categories? Therefore, the contribution of this article is difficult to evaluate.

5. The prompt generator network proposed in this article is based on a simple attention mechanism, which is too simple. Using such a network, why can the generated prompt vectors be generalized to unknown categories?

[1] Improving Zero-shot Generalization and Robustness of Multi-modal Models
[2] Improving Zero-Shot Generalization for CLIP with Synthesized Prompts
[3] SuS-X: Training-Free Name-Only Transfer of Vision-Language Models
[4] Class-aware visual prompt tuning for vision-language pre-trained model
[5] Prompt learning with optimal transport for vision-language models
[6] A Retrospect to Multi-prompt Learning across Vision and Language

**Questions:**

1. What scenarios require training vision-language models under the framework of federated learning?
2. What is the difference between the proposed FedTPG with existing generative prompt learning methods and cross-modal prompt learning methods?
3. Why can the generated prompt vectors be generalized to unknown categories using a simple attention network?

---

> ### Author Response · Authors · 2023-11-21
> **Clarification regarding federated learning setup**
>
> Thank you Reviewer RTLu for the insightful and valuable comments. Below are our responses to your concerns and questions.
>
> **Clarification regarding federated learning setup:**
>
> Our primary objective is to enhance the performance of the CLIP model across a broad spectrum of tasks, datasets and domains. In centralized learning scenarios, obtaining access to the extensive range of tasks can be prohibitively costly, both in terms of data collection and the potential violation of data privacy regulations. Federated learning offers a strategic solution to these challenges by enabling collaborative training of the prompt generator through decentralized communication of training information, instead of private data. This approach not only alleviates data privacy concerns but also reduces the logistical and financial burdens associated with centralized data collection.
>
> For your question about testing on unknown datasets, we utilize this methodology to assess the model's generalization capabilities. Note that the ability to perform effectively on unseen/out-of-distribution datasets/domains is a critical benchmark in current research on CLIP and CLIP-based prompting methods, both in centralized and distributed frameworks. This is evident in existing literature, such as the studies by Guo et al. (2023)[a] and Zhou et al. (2022)[b], which underscore the importance of this metric. Moreover, by testing on unknown datasets, we are not only benchmarking our model's generalization ability but also advancing the broader objective of enhancing CLIP's utility in diverse and practical scenarios.
>
> [a]: Tao Guo, Song Guo, and Junxiao Wang. 2023. PFedPrompt: Learning Personalized Prompt for Vision-Language Models in Federated Learning. In Proceedings of the ACM Web Conference 2023 (WWW '23). Association for Computing Machinery, New York, NY, USA, 1364–1374.
>
> [b] Zhou et al. "Conditional prompt learning for vision-language models." CVPR 2022.

---

> ### Author Response · Authors · 2023-11-21
> **Clarification regarding prompt learning related works [1/2]**
>
> We first thank the reviewer for sharing these references. We added them in the revised version.  Different than learning one model for one task in all prior works, we focus on learning one unified model generalizes well to all seen and unseen tasks with task-related text inputs. Moreover, our method is orthogonal to many of them which makes it straightforward to combine them together for further benefit. We provide the difference and the comparison on generalization ability between our FedTPG to each work as follows:
>
> 1. **Improving Zero-shot Generalization and Robustness of Multi-modal Models**: this work focus on improving the zero-shot generalization ability of CLIP.
> This work uses the WordNet hierarchy; semantics from the parent and child labels to improve prediction accuracy. While this work aims to improve the generalization ability of CLIP with prompt engineering, FedTPG tackles the problem of federated prompt learning. FedTPG has a focus on improving the performance of the few-shot finetuned CLIP models rather than prompt engineering.
>
> 2. The concurrent ICCV 2023 work **Improving Zero-Shot Generalization for CLIP with Synthesized Prompts**: introduces SHIP to improve CLIP’s performance by synthesizing prompts and image features with a trained variational autoencoder for categories lacking data. SHIP requires a retraining process on the synthesized image features for each new classification problem, while FedTPG generates context-aware prompt vectors with one forward pass, and therefore is more efficient at inference time. Our proposed method is also compatible with utilizing synthesized image features. However, the focus of our paper is studying the effect of text-conditional prompt generation in federated prompt learning, which is orthogonal to the focus of this work.
>
> 3. The concurrent ICCV 2023 work **SuS-X: Training-Free Name-Only Transfer of Vision-Language Models**: has a different concentration than prompt learning (and federated learning). It proposes to construct a support image dataset given the target class names. It improves CLIP prediction by taking into account both text inputs and the image features from the constructed support set. Our proposed method is also compatible with utilizing additional images from the constructed support set. However, the focus of our paper is studying the effect of text-conditional prompt generation in federated prompt learning, which is orthogonal to this work.
>
> 4. **Class-aware visual prompt tuning for vision-language pre-trained model**: CAVPT proposes prompt learning on both the text and the visual modalities simultaneously. Beyond text and visual prompting, CAVPT provides a class-aware visual prompt to make the vision prompt concentrate on the target class concept. Our FedTPG algorithm can be extended to CAVPT by learning multiple prompt generators for each modality. CAVPT is orthogonal to FedTPG, which focuses more on individual target domains instead of the generalization ability across different domains.
>
> 5. **Prompt learning with optimal transport for vision-language models**: PLOT is an extension to CoOp, instead of a single prompt, PLOT optimizes for multiple prompts to describe different characteristics of a category. Our context conditioned text-driven prompt generation approach can also be extended to PLOT either by learning a single prompt generator network to produce multiple prompts or learning multiple prompt generator networks each produce a prompt using the PLOT optimization technique. Our method is orthogonal to PLOT and based on the results shown in our paper, we believe that our context conditioning approach further benefits PLOT.
>
> 6. The concurrent ICCV 2023 work **A Retrospect to Multi-prompt Learning across Vision and Language (EMPL)**: This work also utilizes multiple prompt templates that are obtained by drawing prompt instances from an underlying Energy Based Model(EBM) conditional prompt distribution. They meta-learn unseen classes across different tasks during training and optimizes for generic prompt learning goal (as in CoOp) and EBM uncertainty modeling.
> Their optimization objective can directly be used for training our text-driven prompt generation network. Therefore, our text-driven prompt generation approach can provide better contextual prompts on unseen classes that can be complemented with EMPL[6] generalization ability during inference.

---

> ### Author Response · Authors · 2023-11-21
> **Clarification regarding prompt learning related works [2/2]**
>
> We appreciate the reviewer's suggestion regarding the comparison of FedTPG with other existing prompt learning methods. We concur that such comparisons are valuable in demonstrating the effectiveness of FedTPG and thank you for pointing out these related works. Due to our limited computational budget, we are unable to provide the empirical comparisons within the given time frame. As explained above, one major and common difference to all prior works is that our approach learns one unified model for base/new classes of all datasets, while all prior works learn one model for one dataset. We study the impact of using text-driven prompt generation for generalizing prompt learning to many different tasks scalably, which is underexplored in all prior works. We realize that these prior works are orthogonal and can be complementary to our proposed text-conditional prompt generation technique. We believe that combining them with our method would further improve performance.
>
> To address the need for relevant comparisons, we have incorporated comparative analyses of FedTPG with FedCoOp, FedKgCoOP and two more existing prompt learning strategies: CoCoOp and Maple. In the context of federated learning, these are adapted as FedCoCoOp and FedMaple, respectively. We have added the experimental results in Table1-3 at the main paper with analysis at Pages 7-8. In this response, we attached the averaged performance at Table 1-a, 2 and 3 as follows:
>
> | Method    | Local | Base  | New   | HM    |
> |--------------|-------|-------|-------|-------|
> | FedCoCoOp    | 81.41 | 73.87 | 65.98 | 73.21 |
> | FedMaple     | **81.63** | **74.44** | 70.59 | 75.28 |
> | FedTPG       | 80.75 | 73.68 | **76.02** | **76.70** |
>
> *Table 1 (a) Average over 9 datasets, Base to New*
>
> | Method     | ImageNet | Average (ImageNet variants) | Average (10 unseen Datasets) |
> |-----|-----|----------------------|----------------------|
> | FedCoCoOp  | 68.51    | 59.00             | 64.50             |
> | FedMaple   | 66.96    | 56.55             | 62.98             |
> | FedTPG     | **69.51**    | **59.22**             | **65.31**             |
>
> *Zero-shot transfer from ImageNet to its variants and 10 datasets.*
>
> From the observation, it is evident that FedTPG consistently outperforms FL variants of existing advanced prompt learning methods, including FedMaple and FedCoCoOp in terms of New and Harmonic Mean (HM) in the base to new benchmark, and zero-shot transfer ability in both the ImageNet variants and the 10 datasets.

---

> ### Author Response · Authors · 2023-11-21
> **Thanks for your valuable comments**
>
> **Clarification regarding Table 1:**
>
> As shown in Table 1 and further substantiated in Tables 6 and 7 where each client has fewer accessible classes, the proposed FedTPG consistently demonstrates the highest averaged Harmonic Mean (HM) in base-to-new experiments, even under conditions of increased heterogeneity (where each client handles fewer classes). This consistently high HM indicates that FedTPG simultaneously advances performance across the three pivotal benchmarks in the paper: "Local" performance for each client's unique dataset, shared "Base" performance across the federated learning (FL) network, and robust generalization to "New"(out-of-distribution) datasets or classes.
>
> Importantly, these advantages of FedTPG are not achieved at the sacrifice of performance in known categories. Rather than compromising performance on local and base classes, FedTPG enhances the efficacy of CLIP in these areas without degrading its ability to generalize to new datasets. For example, FedTPG offers a comprehensive improvement over the public CLIP across local, base, and new classes. This is in contrast to approaches like FedCoOp, which, while achieving higher performance on base classes, do so by sacrificing accuracy in new classes compared to CLIP.
> Therefore, our approach represents a balanced and effective advancement in federated learning, contributing to both specific and generalized performance improvements without detriment to either aspect.
>
> **Clarification regarding its generalization ability:**
>
>  The advantages of proposed FedTPG can be summarized to two primary components: (1) the task-related text inputs, and (2) the attention module.
>
> 1. **Task-Related Text Inputs**: One main contribution of our method is utilizing task-related text inputs for context-aware prompt generation. The inputs are crucial in providing compact yet rich context semantics pertinent to the target problem, enabling the generalizability of the  context-aware prompt vectors in FedTPG. This cross-domain context awareness is fundamental to the generated prompts' ability to generalize effectively to unknown tasks.
>
> 2. **Attention Module**: The attention module in FedTPG acts as a function summarizing uneven lengths of text inputs and extract essential context information. Specifically, it takes the input text embeddings of all available class names and  efficiently maps them to the desired input token space of the CLIP model. The simplicity of the attention mechanism does not detract from its effectiveness. Instead, it provides a focused and efficient way to process and summarize the text embeddings, Moreover, this module is intentionally designed to be lightweight to minimize communication costs in a federated learning setup.
>
> In summary, the combination of task-relevant text inputs and an attention module enables FedTPG to generate prompt vectors that are both contextually rich and generalizable. This is achieved while maintaining a low communication overhead, crucial in federated learning environments.

---

### Official Review · Reviewer_qkce · 2023-10-28

**Soundness:** 3 good
**Presentation:** 3 good
**Contribution:** 2 fair
**Rating:** 6
**Confidence:** 3

**Summary:**

Prompt learning for vision-language models has achieved remarkable success in the adaptation of CLIP to different downstream tasks. This paper considers apply this approach to the field of federated learning. Specifically, they present Federated Text-driven Prompt Generation (FedTPG), which enables the scalable learning of a unified prompt generation network across multiple remote clients. Since the prompt generation network is conditioned on task-related text input, it is context-aware and suitable for generalization to both seen and unseen classes. The comprehensive empirical evaluations on nine diverse image classification datasets demonstrate that the proposed method outperforms existing federated prompt learning methods.

**Strengths:**

1.	This is the first paper that solves the challenge of using prompt learning in FL for unseen classes, which is of great importance.
2.	The idea of adopting a prompt generator for context-aware tasks is novel.
3.	The experimental results demonstrate the effectiveness of the proposed method.

**Weaknesses:**

1.	The computational cost is high. Although the CLIP pre-trained model is fixed during local training, it still requires backward propagation over the encoder. Considering the clients may be edge devices with limited resources, this may hinder the wide adoption of the proposed method.
2.	The generator is more likely designed for the centralized settings instead of the decentralized settings. It appears to be a simple prompt strategy in the centralized setting. Distributed properties and NonIID data distribution are not well taken into account. As a consequence, many existing prompt learning strategies for centralized settings may also be directly adopted and should be compared.

**Questions:**

no

---

> ### Author Response · Authors · 2023-11-21
> **Thanks for your positive and valuable comments**
>
> Thank you Reviewer qkce for the positive and valuable comments. Below are our responses to your concerns.
>
> **Clarification regarding computation cost:**
>
> The requirement for backward propagation over the CLIP text encoder holds true for the gradient based prompt learning problem setup, which encompasses all the existing prompt learning prior works (CoOp, CoCoOp, Maple, KgCoOp). Our work falls in the same line of direction on gradient based prompt learning, but in a federated learning setting. For edge devices, substituting the standard CLIP model with a mobile-friendly version, such as distilled version of CLIP (https://tech.pic-collage.com/distillation-of-clip-model-and-other-experiments-f8394b7321ce) can reduce the computational cost significantly, and therefore our method FedTPG can be readily adapted to such compact CLIP model.
>
> Though the proposed FedTPG involves gradient computation through the frozen CLIP layers (e.g., encoder), the communication cost can be very lightweight: only the compact prompt generator with a lightweight cross-attention module is transferred among the federated learning distributed network. Note that different from centralized learning that bounded by the model training, the bottleneck of FL network is typically the communication cost. From this perspective, our design can significantly reduce the communication overhead, which is a primary concern, resulting considerably low communication cost and making the process efficient and manageable.
>
> **Comparing with additional FL-variants of  prompt learning baselines:**
>
> Our work focuses a federated learning scenario, which simulates a real-world application of prompting foundational models like CLIP. For example, in the healthcare and finance field, due to the sensitive nature of data and the regulatory constraints on data sharing, accessing and aggregating multiple datasets for conventional prompt learning can be impractical. In these sectors, distributed learning becomes not just a beneficial strategy, but a necessary one, where the learned information of prompting public available CLIP with private data can be shared for further generalization purpose. Hence, the constructed FL network in the paper involves numerous clients, each possessing distinct and heterogeneous datasets from different domains.
>
> Our primary goal is to develop one unified prompt generator capable of delivering enhanced performance across both familiar (seen in the network) and novel (out-of-distribution) classification problems. The proposed FedTPG algorithm is designed to navigate both challenges of federated learning and prompt learning, enabling the leveraging of diverse data sources while maintaining the integrity and confidentiality of each client's dataset.
>
> Meanwhile, we have added the comparison of FedTPG with two additional prompt learning strategies CoCoOp and MaPLe under the federated learning setting, called FedCoCoOp and FedMaPLe. We have added the experimental results in Table1-3 in the main paper. In this response, we attached the averaged performance at Table 1-a, 2 and 3 as follows:
>
> | Method    | Local | Base  | New   | HM    |
> |--------------|-------|-------|-------|-------|
> | FedCoCoOp    | 81.41 | 73.87 | 65.98 | 73.21 |
> | FedMaPLe     | **81.63** | **74.44** | 70.59 | 75.28 |
> | FedTPG       | 80.75 | 73.68 | **76.02** | **76.70** |
>
> *Table 1 (a) Average over 9 datasets, Base to New*
>
> | Method     | ImageNet | Average (ImageNet variants) | Average (10 unseen Datasets) |
> |-----|-----|----------------------|----------------------|
> | FedCoCoOp  | 68.51    | 59.00             | 64.50             |
> | FedMaPLe   | 66.96    | 56.55             | 62.98             |
> | FedTPG     | **69.51**    | **59.22**             | **65.31**             |
>
> *Zero-shot transfer from ImageNet to its variants and 10 datasets.*
>
> From the observation, it is evident that FedTPG achieves the highest Harmonic Mean (HM) over local, base, and new benchmarks. Specifically, FedTPG outperforms FedCoCoOp and FedMaPLe on the new classes significantly and achieves compatible performance on the local and base classes.
>
> Furthermore, in the context of ImageNet training and zero-shot transfer evaluation, our proposed FedTPG model surpasses both FedCoCoOp and FedMaPLe. Such findings underscore the advantages of FedTPG over centralized prompt learning techniques, particularly in addressing the challenges posed by non-iid and distributed properties inherent in Federated Learning.

---

> > ### Comment · Reviewer_qkce · 2023-11-23
> > **Response to authors**
> >
> > Thanks for your response. I have carefully read the responses. I agree that the compact CLIP model may be suitable for edge devices. Yet, this will also limit the application scenarios of the algorithm. Besides, the improvement of the algorithm is small. In some metrics, it is not even better than the baseline. Considering these, I decided to keep my score unchanged.

---

### Official Review · Reviewer_o1Q3 · 2023-10-30

**Soundness:** 2 fair
**Presentation:** 3 good
**Contribution:** 2 fair
**Rating:** 6
**Confidence:** 3

**Summary:**

This paper propose a Federated prompt tuning method, where a unified prompt generation network shared by all clients are proposed to generate specific prompt for each client based on the dataset context. The experiments show that the proposed method outperforms existing federated prompt tuning methods on both based-to-new and dataset generalization task.

**Strengths:**

1. The motivation of a context-aware prompt generation network is reasonable.
2. The experiments show that the proposed methods outperform existing federated prompt tuning methods.

**Weaknesses:**

1. Can the author provide a more detailed analysis on the different between the proposed method and CoCoOp, since it also introduces an extra prompt generalization network.
2. Like CoCoOp, the image may also provide important context informatio. How is the performance of proposed model compared to a model taking image as input? Will using both visual and textual information as context further improve the performance?
3. The proposed method does not achieve the state-of-the-art performance on 3 out of 8 datasets in terms of the base-to-new task. Can the authors provide a more detailed analysis on the characteristics of these 3 datasets and why does the proposed method performs relatively worse on these datasets?

**Questions:**

Please refer to the weaknesses.

---

> ### Author Response · Authors · 2023-11-21
> **Comparing with CoCoOp under federated learning setup**
>
> Thank you Reviewer o1Q3 for the positive and valuable comments. Below are our responses to your concerns.
>
> **1. Comparing with CoCoOp:**
>
> Compared with CoCoOp, FedTPG generates context-aware prompt vectors conditioning on the task-related text inputs. This approach leverages the compact nature of text to store context semantics, which is more efficient than using images in CoCoOp. Typically, images contain more extraneous information, raising the complexity in extracting the characteristic features for prompt generation.
>
> We provide comparisons to CoCoOp under the federated learning settings, called FedCoCoOp in table 1,2, and 3 in the revised version. In this response, we attach the comparisons of averaged performance in base-to-new and the ImageNet zero-shot transfer settings as follows (please refer to the revised main paper for more detail.):
>
> | Method    | Local | Base  | New   | HM    |
> |--------------|-------|-------|-------|-------|
> | FedCoCoOp    | **81.41** | **73.87** | 65.98 | 73.21 |
> | FedTPG       | 80.75 | 73.68 | **76.02** | **76.70** |
>
> *Table 1 (a) Average over 9 datasets, Base to New*
>
> | Method     | ImageNet | Average (ImageNet variants) | Average (10 unseen Datasets) |
> |-----|-----|----------------------|----------------------|
> | FedCoCoOp  | 68.51    | 59.00             | 64.50             |         |
> | FedTPG     | **69.51**    | **59.22**             | **65.31**             |
>
> *Zero-shot transfer from ImageNet to its variants and 10 datasets.*
>
> These comparisons demonstrate that FedTPG, which utilizes text for prompt generation, consistently outperforms FedCoCoOp on the two considered benchmarks, especially on the generalization ability.
>
> **2. Comparing with baselines using both visual and textual information:**
>
> We thank the reviewer for providing this suggestion. We agree with the reviewer that fusing both visual and textual information could potentially enhance model performance. We consider that this aspect falls outside the scope of our current study and leads to a potential research direction for further investigation.
>
> Meanwhile, we also compare with another recent work which focuses on the prompt learning on both the image and text modalities, *MaPLe: Multi-modal Prompt Learning*. Similar to CoCoOp, we implement Maple under our federated learning setup, named as FedMaple, where the compared results are highlighted as follows:
>
>
> | Method    | Local | Base  | New   | HM    |
> |--------------|-------|-------|-------|-------|
> | FedMaple     | **81.63** | **74.44** | 70.59 | 75.28 |
> | FedTPG       | 80.75 | 73.68 | **76.02** | **76.70** |
>
> *Table 1 (a) Average over 9 datasets, Base to New*
>
> | Method     | ImageNet | Average (ImageNet variants) | Average (10 unseen Datasets) |
> |-----|-----|----------------------|----------------------|
> | FedMaple   | 66.96    | 56.55             | 62.98             |
> | FedTPG     | **69.51**    | **59.22**             | **65.31**             |
>
> *Zero-shot transfer from ImageNet to its variants and 10 datasets.*
>
> The above comparison results demonstrate that FedTPG outperforms FedMaple with only prompting the text context.
>
>
> **3. Analysis on OxfordPets, DTD, and Food102:**
>
> In our analysis, as detailed in Appendix Table 5, the OxfordPets and DTD datasets comprise 37 and 47 classes, respectively. With the experimental setting that each client handling approximately 20 classes in Table 1, the base classes for either OxfordPets or DTD are exclusively managed by a single client during the federated learning model training process. It's important to note that these local classes are precisely the same as the base classes.
>
> In this scenario, the local-training baseline, CoOp, is trained on "Base" classes for "Base" evaluation, where the model does not encounter the non-IID challenges typically associated with Federated Learning (FL). Consequently, the superior performance on the "Local" classes in CoOp is also demonstrated for the "Base" classes that our proposed FedTPG does not achieve the best. Note that these findings are underscored by our results in Appendix Table 6 and Table 7, where we explore scenarios with each client handling a reduced number of classes (either 5 or 10). In these cases, FedTPG consistently outperforms other methods for the OxfordPets and DTD datasets.
>
> Regarding the Food102 dataset, we observe a minimal performance discrepancy among the various methods. Despite these marginal differences, FedTPG consistently delivers competitive results, showcasing its robustness across different datasets and domain distributions.

---

### Official Review · Reviewer_KFdC · 2023-11-02

**Soundness:** 2 fair
**Presentation:** 3 good
**Contribution:** 2 fair
**Rating:** 6
**Confidence:** 3

**Summary:**

Prompt learning for vision-language models has shown promising results in adapting CLIP to different downstream tasks. The paper proposes to improve the generalizability of the learn prompts to unseen classes, by proposing to solve this problem in a federated learning setting. Specifically, the prompt generation network is conditioned on task-related text embeddings, making the generation process context-aware and generalizes to the unseen classes. The paper evaluates the proposed approach on a diverse set of 9 datasets, as well as different ImageNet variants and demonstrates better generalization to unseen classes and domains than the baseline CLIP and CoOp.

**Strengths:**

- The generalization of the prompt tuning vision-language models is an important research topic, and the federated learning setting is interesting.
- The proposed technique effectively improves the generalizability of the learned prompts.
- The paper is well written.

**Weaknesses:**

- What is the reason of not comparing to CoOp on EuroSAT/ImageNet in Table 1? This seems to be the setting followed by many works along this direction.

- What is the reason of not comparing to CoCoOp [1] and MaPLe [2] in the base-to-new generalization setting? Will the proposed technique have the same benefit if naively applied to Fed-CoCoOp / Fed-MaPLe?

- Why are the baseline numbers of CLIP/CoOp different between Table 1 of this paper and CoCoOp Table 1? Is it because the different selection of the base class and why?

[1] Zhou et al. "Conditional prompt learning for vision-language models." CVPR 2022.
[2] Khattak et al. "Maple: Multi-modal prompt learning." CVPR 2023.

**Questions:**

Please see the weaknesses section.

---

> ### Author Response · Authors · 2023-11-21
> **Addressing concerns on baseline comparisons [1/2]**
>
> We thank Reviewer KFdC for the constructive and valuable comments. Below are our responses to your concerns and questions.
>
> **1. Benchmarking on ImageNet/EuroSAT:**
>
> We would like to emphasize that, our work primarily operates in Federated Learning (FL) setup. Compare to prior works on prompt learning in FL, such as Guo et al. (2023a)[a] and Guo et al. (2023b)[b], we report results on more datasets, and the 9 datasets in Table~1 is the most comprehensive experimental setup in FL setting currently.
>
> Missing EuroSAT in Table 1: As mentioned in our Experimental setup under Sec 4.1, we considered for each client in FL setup to own 20 classes of all the nine datasets. Note that this choice of 20 classes is arbitrary and Table 4 shows the ablation study of this choice. However, EuroSAT contains a total of 10 classes. To maintain a consistent number of classes from each dataset i.e, 20 classes, we excluded EuroSAT as it's total number of classes is less than our choice of 20 classes and and therefore does not support our federated experimental setup.
>
> Missing ImageNet in Table 1: We aim to learn one unified model across all the datasets, while all prior prompt learning works, e.g., CoOp, CoCoOp, Maple, learn an independent model for each dataset. In table 1, we show that FedTPG learns one unified model on medium sized datasets and improves performance on all those datasets. If we include ImageNet in table 1, it is hard to deduce whether the generalization ability is derived from the large-scale ImageNet or other medium size datasets.
>
> In table 2 and 3, we provide the federated prompt learning evaluation on ImageNet, which corresponds to the centralized prompt learning evaluation on ImageNet in prior works. Since we know ImageNet is a strong data source, we treat ImageNet as an independent testbed to evaluate if FedTPG can learn one model on ImageNet itself that generalizes to its variants in Table 2, and to 10 unseen datasets (including **EuroSAT**) in table 3.
>
>
> **2. Comparison with CoCoOp and MaPle:**
>
> We thank you for providing this suggestion. Comparisons with FedCoCoOp and FedMaple now can be found in Table 1-3 in our manuscript. Below, we report the average performance of FedCoOp, FedMaPle, and FedTPG in Table 1 as follows:
>
> |              | Local | Base  | New   | HM    |
> |--------------|-------|-------|-------|-------|
> | FedCoCoOp    | 81.41 | 73.87 | 65.98 | 73.21 |
> | FedMaple     | **81.63** | **74.44** | 70.59 | 75.28 |
> | FedTPG       | 80.75 | 73.68 | **76.02** | **76.70** |
>
> *Average over 9 datasets.*
>
> This comparison shows that FedTPG improves the performance on the new classes significantly and have a comparable performance on the local and base classes. FedTPG achieves the highest HM averaging over local, base, and new experiments, demonstrating its balanced improvements on various cases.
>
> The zero-shot transfer from ImageNet to 10 datasets of FedCoOp, FedMaPle, and FedTPG in Table 3 is as follows:
>
> |           | ImageNet | Caltech101 | Flowers102 | FGVCAircraft | UCF101 | OxfordPets | Food101 | DTD   | StanfordCars | SUN397 | EuroSAT | Average |
> |------|------|--------|--------|---------|--------|------------|---------|-------|--------------|--------|---------|---------|
> | FedCoCoOp | 68.51    | **94.11**      | 66.34      | 20.79        | 62.75  | 89.04      | 85.40   | 43.20 | 63.98        | 64.02  | **55.40**   | 64.50   |
> | FedMaple  | 66.96    | 92.49      | 68.25      | **23.52**        | 60.32  | 89.67      | 83.52   | 44.68 | 60.16        | 61.85  | 45.38   | 62.98   |
> | FedTPG    | **69.51** | 93.75  | **70.04**      | 23.22        | **64.72** | **90.60**  | **85.91**   | **46.25** | 63.98        | **66.78** | 47.86| **65.31** |
>
> *Zero-shot transfer from ImageNet to 10 datasets.*
>
> From the experimental observation, we notice that our proposed FedTPG achieves better performance at ImageNet and zero-shot transfer to the 10 datasets against FedCoCoOp and FedMaple.

---

> > ### Author Response · Authors · 2023-11-21
> > **Addressing concerns on baseline comparisons [2/2]**
> >
> > **3. Different experimental setup than CoCoOp:**
> >
> > The selected base and new classes in our paper and CoCoOp [1] are the same.  The reason is that the experimental setup details between our paper and the CoCoOp paper are different.
> >
> > 1. In our study, we consider a federated training setup that differs significantly from the training setup presented in the prompt learning papers, e.g., CoOp, CoCoOp, Malpe. The federated training setup simulates a real-world distributed training environment, where the training data is distributed to many clients. Each client has access to a limited number of data classes. In case we have 50 classes distributed to 5 clients for training, CoOp trained on a local client handling 10 classes works well on the local 10 classes (**Local Performance**), but cannot generalize to the 50 base classes (**Base Performance**) and  new classes (**New Performance**). FedCoOp trained on 5 clients collaboratively achieves better results on the 50 base classes, but still suffers from generalizing to new classes. In summary, the evaluation of the base classes is a harder task than it in the prompt learning papers due to our practical federated learning setup. Notice in table 1, FedTPG learns one unified model that solves the classification tasks on all nine datasets.
> >
> > 2. For CLIP, note that the results in our paper are averaged over three runs with different random seeds. The different seeds can cause a minor variation, even though the CLIP model is frozen during training. However, the averaged results in our paper on CLIP are very close to the results in the CoCoOp paper.
> >
> > [a]: Tao Guo, Song Guo, and Junxiao Wang. 2023. PFedPrompt: Learning Personalized Prompt for Vision-Language Models in Federated Learning. In Proceedings of the ACM Web Conference 2023 (WWW '23). Association for Computing Machinery, New York, NY, USA, 1364–1374.
> > [b]: T. Guo, S. Guo, J. Wang, X. Tang and W. Xu, "PromptFL: Let Federated Participants Cooperatively Learn Prompts Instead of Models - Federated Learning in Age of Foundation Model," in IEEE Transactions on Mobile Computing.

---

> > > ### Comment · Reviewer_KFdC · 2023-11-23
> > >
> > > I thank the authors for the clarification on the experiment settings and for providing the additional results. Most of my concerns are addressed and I am happy to increase my rating.

---

### Meta-Review · Area_Chair_UquK · 2023-12-05

**Metareview:**

This paper propose a Federated prompt tuning method, where a unified prompt generation network shared by all clients is used to generate specific prompt for each client based on the dataset context. The experiments show that the proposed method outperforms existing federated prompt tuning methods on both based-to-new and dataset generalization task.

**Justification For Why Not Higher Score:**

Results are not always SOTA when including comparisons to more recent prompt learning methods like maple and Co-CoOp, similarly results lag behind on base-to-new

**Justification For Why Not Lower Score:**

the rebuttal addressed concerns remaining from the reviewers

---

### Decision · Program_Chairs · 2024-01-16

Accept (poster)